# EDA: Easy Data Augmentation Techniques for Boosting Performance on Text Classification Tasks

**Jason Wei**[1,2]    **Kai Zou**[3]
[1]Protago Labs Research, Tysons Corner, Virginia, USA
[2]Department of Computer Science, Dartmouth College
[3]Department of Mathematics and Statistics, Georgetown University
jason.20@dartmouth.edu    kz56@georgetown.edu

## Abstract

We present **EDA**: **e**asy **d**ata **a**ugmentation techniques for boosting performance on text classification tasks. EDA consists of four simple but powerful operations: synonym replacement, random insertion, random swap, and random deletion. On five text classification tasks, we show that EDA improves performance for both convolutional and recurrent neural networks. EDA demonstrates particularly strong results for smaller datasets; on average, across five datasets, training with EDA while using only 50% of the available training set achieved the same accuracy as normal training with all available data. We also performed extensive ablation studies and suggest parameters for practical use.

## 1 Introduction

Text classification is a fundamental task in natural language processing (NLP). Machine learning and deep learning have achieved high accuracy on tasks ranging from sentiment analysis (Tang et al., 2015) to topic classification (Tong & Koller, 2002), but high performance is often dependent on the size and quality of training data, which is often tedious to collect. Automatic data augmentation is commonly used in vision (Simard et al., 1998; Szegedy et al., 2014; Krizhevsky et al., 2017) and speech (Cui et al., 2015; Ko et al., 2015) and can help train more robust models, particularly when using smaller datasets. However, because it is difficult to come up with generalized rules for language transformation, universal data augmentation techniques in NLP have not been explored.

Previous work has proposed techniques for data augmentation in NLP. One popular study generated new data by translating sentences into French and back into English (Yu et al., 2018). Other works have used predictive language models for synonym replacement (Kobayashi, 2018) and data noising as smoothing (Xie et al., 2017). Although these techniques are valid, they are not often used in practice because they have a high cost of implementation relative to performance gain.

In this paper, we present a simple set of universal data augmentation techniques for NLP called EDA (**e**asy **d**ata **a**ugmentation). To the best of our knowledge, we are the first to comprehensively explore text editing techniques for data augmentation. We systematically evaluate EDA on five benchmark classification tasks, and results show that EDA provides substantial improvements on all five tasks and is particularly helpful for smaller datasets. Code will be made publicly available.

| Operation | Sentence |
|-----------|----------|
| None | A sad, superior human comedy played out on the back roads of life. |
| SR | A *lamentable*, superior human comedy played out on the *backward* road of life. |
| RI | A sad, superior human comedy played out on *funniness* the back roads of life. |
| RS | A sad, superior human comedy played out on *roads* back *the* of life. |
| RD | A sad, superior human out on the roads of life. |

Table 1: Sentences generated using EDA. SR: synonym replacement. RI: random insertion. RS: random swap. RD: random deletion.

## 2 EDA

Frustrated by the measly performance of text classifiers trained on small datasets, we tested a number of augmentation operations loosely inspired by those used in vision and found that they helped train more robust models. Here, we present the full details of EDA. For a given sentence in the training set, we perform the following operations:

1. **Synonym Replacement (SR):** Randomly choose $n$ words from the sentence that are not stop words. Replace each of these words with one of its synonyms chosen at random.
2. **Random Insertion (RI):** Find a random synonym of a random word in the sentence that is not a stop word. Insert that synonym into a random position in the sentence. Do this $n$ times.
3. **Random Swap (RS):** Randomly choose two words in the sentence and swap their positions. Do this $n$ times.
4. **Random Deletion (RD):** Randomly remove each word in the sentence with probability $p$.

Since long sentences have more words than short ones, they can absorb more noise while maintaining their original class label. To compensate, we vary the number of words changed, $n$, for SR, RI, and RS based on the sentence length $l$ with the formula $n=\alpha\,l$, where $\alpha$ is a parameter that indicates the percent of the words in a sentence are changed (we use $p=\alpha$ for RD). Furthermore, for each original sentence, we generate $n_{aug}$ augmented sentences. Examples of augmented sentences are shown in Table 1. We note that synonym replacement has been used previously (Kolomiyets et al., 2011; Zhang et al., 2015; Wang & Yang, 2015), but to our knowledge, random insertions, swaps, and deletions have not been studied.

## 3 EXPERIMENTS

### 3.1 EXPERIMENTAL SETUP

We conduct experiments on five benchmark text classification tasks: (1) **SST-2**: Stanford Sentiment Treebank (Socher et al., 2013), (2) **CR**: customer reviews (Hu & Liu, 2004; Liu et al., 2015), (3) **SUBJ**: subjectivity/objectivity dataset (Pang & Lee, 2004), (4) **TREC**: question type dataset (Li & Roth, 2002), and (5) **PC**: Pro-Con dataset (Ganapathibhotla & Liu, 2008). Summary statistics are shown in Table 3 in the Appendix. Furthermore, we hypothesize that EDA is more helpful for smaller datasets, so we delegate the following sized datasets by selecting a random subset of the full training set with $N_{train}=\{500, 2{,}000, 5{,}000, \text{all available data}\}$.

We run experiments for two state-of-the-art models in text classification. **(1)** Recurrent neural networks (RNNs) are suitable for sequential data. We use a LSTM-RNN (Liu et al., 2016). **(2)** Convolutional neural networks (CNNs) have also achieved high performance for text classification. We implement them as described in (Kim, 2014). Details are in Section 6.1 in the Appendix.

### 3.2 EDA MAKES GAINS

We run both CNN and RNN models with and without EDA across all five datasets for varying training set sizes. Average performances (%) are shown in Table 2. Of note, average improvement was 0.8% for full datasets and 3.0% for $N_{train}=500$.

|  | Training Set Size | | | |
| --- | --- | --- | --- | --- |
| Model | 500 | 2,000 | 5,000 | full set |
| RNN | 75.3 | 83.7 | 86.1 | 87.4 |
| +EDA | 79.1 | 84.4 | 87.3 | 88.3 |
| CNN | 78.6 | 85.6 | 87.7 | 88.3 |
| +EDA | 80.7 | 86.4 | 88.3 | 88.8 |
| *Average* | 76.9 | 84.6 | 86.9 | 87.8 |
| +EDA | 79.9 | 85.4 | 87.8 | **88.6** |

Table 2: Average performances (%) across five text classification tasks for models without and without EDA on different training set sizes.

Overfitting tends to be more severe when training on smaller datasets. By conducting experiments using a restricted fraction of the available training data, we show that EDA has more significant improvements for smaller training sets. We run both normal training and EDA training for the following training set fractions (%): $\{1, 5, 10, 20, 30, 40, 50, 60, 70, 80, 90, 100\}$. Figure 1 shows average performance across all datasets. The best average accuracy without augmentation, 88.3%,

was achieved using 100% of the training data. Models trained using EDA surpassed this number by achieving an average accuracy of 88.6% while only using 50% of the available training data. Results for individual datasets are displayed in Figure 4 (Appendix).

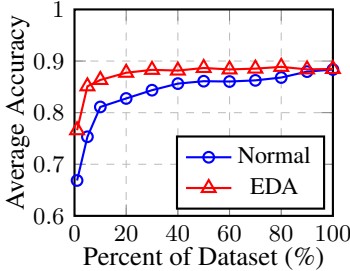

Figure 1: Performance on text classification tasks with respect to percent of dataset used for training.

### 3.3 CONSERVING TRUE LABELS

In data augmentation, input data is altered while class labels are maintained. However, if sentences are significantly changed, then original class labels may no longer be valid. We take a visualization approach to examine whether EDA operations significantly change the meanings of augmented sentences. First, we train an RNN on the pro-con classification task (PC) without augmentation. Then, we apply EDA to the test set by generating nine augmented sentences per original sentence. These are fed into the RNN along with the original sentences, and we extract the outputs from the last dense layer. We apply t-SNE (Van Der Maaten, 2014) to these vectors and plot their 2-D representations (Figure 2). We found that the resulting latent space representations for augmented sentences closely surrounded those of the original sentences.

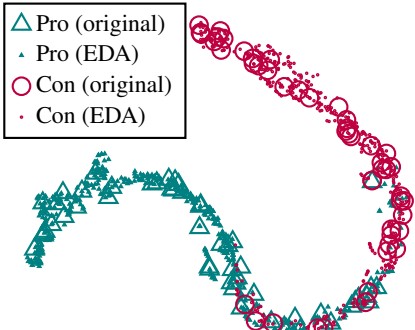

Figure 2: Latent space visualization of original and augmented sentences in the Pro-Con dataset.

### 3.4 ABLATION STUDIES

So far, we have shown encouraging empirical results. In this section, we perform ablation studies to explore the effects of each component in EDA. Synonym replacement has been previously used (Kolomiyets et al., 2011; Zhang et al., 2015; Wang & Yang, 2015), but the other three EDA operations have not yet been explored. One could hypothesize that the bulk of EDA's performance gain is from synonym replacement, so we isolate each of the EDA operations to determine their individual ability to boost performance. For all four operations, we ran models using a single operation while varying the augmentation parameter $\alpha=\{0.05, 0.1, 0.2, 0.3, 0.4, 0.5\}$ (Figure 3).

It turns out that all four EDA operations contribute to performance gain. For SR, improvement was good for small $\alpha$, but high $\alpha$ hurt performance, likely because replacing too many words in a sentence changed the identity of the sentence. For RI, performance gains were more stable for different $\alpha$ values, possibly because the original words in the sentence and their relative order were maintained in this operation. RS yielded high performance gains at $\alpha \leq 0.2$, but declined at $\alpha \geq 0.3$ since performing too many swaps is equivalent to shuffling the entire order of the sentence. RD had the highest gains for low $\alpha$ but severely hurt performance at high $\alpha$, as sentences are likely unintelligible if up to half the words are removed. Improvements were more substantial on smaller datasets for all operations, and $\alpha=0.1$ appeared to be a "sweet spot" across the board.

The natural next step is to determine how the number of generated augmented sentences per original sentence, $n_{aug}$, affects performance. We calculate average performances over all datasets for $n_{aug}=\{1, 2, 4, 8, 16, 32\}$, as shown in Figure 3 (middle).

On smaller training sets, overfitting was more likely, so generating many augmented sentences yielded large performance boosts. For larger training sets, adding more than four augmented sentences per original sentence was unhelpful since models tend to generalize properly when large quantities of real data are available. Based on these results, we recommend parameters for practical use in Figure 3 (right).

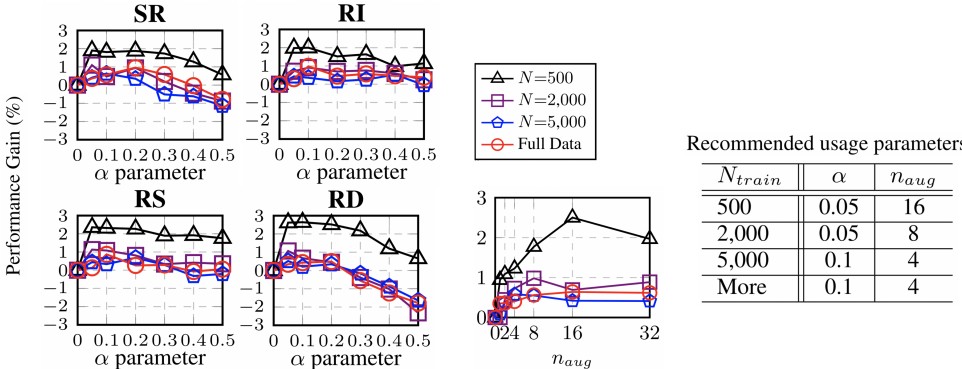

Figure 3: Average performance gain of EDA operations over five text classification tasks for different training set sizes. The $\alpha$ parameter roughly means "percent of words in sentence changed by each augmentation." SR: synonym replacement. RI: random insertion. RS: random swap. RD: random deletion. $n_{aug}$: number of generated augmented sentences per original sentence.

## 4 COMPARISON TO PREVIOUS WORK

Related work is creative but often complex. Back-translation (Sennrich et al., 2016), translational data augmentation (Fadaee et al., 2017), and noising (Xie et al., 2017) have shown improvements in BLEU measure for machine translation. For other tasks, previous approaches include task-specific heuristics (Kafle et al., 2017) and back-translation (Silfverberg et al., 2017; Yu et al., 2018). Regarding synonym replacement (SR), one study showed a 1.4% F1-score boost for tweet classification by finding synonyms with k-nearest neighbors using word embeddings (Wang & Yang, 2015). Another study found no improvement in temporal analysis when replacing headwords with synonyms (Kolomiyets et al., 2011), and mixed results were reported for using SR in character-level text classification (Zhang et al., 2015); however, neither work conducted extensive ablation studies.

Most studies explore data augmentation as a complementary result for translation or in a task-specific context, so it is hard to directly compare EDA to previous literature. But there are two studies similar to ours that evaluate augmentation techniques on multiple datasets. Hu et al. (2017) proposed a generative model that combines a variational auto-encoder (VAE) and attribute discriminator to generate fake data, demonstrating a 3% gain in accuracy on two datasets. Kobayashi (2018) showed that replacing words with other words that were predicted from the sentence context using a bi-directional language model yielded a 0.5% gain on five datasets. However, training a variational auto-encoder or bidirectional LSTM language model is a lot of work. EDA yields similar results but is much easier to use because it does not require training a language model and does not use external datasets. In Table 4 (Appendix), we show EDA's ease of use compared to other techniques.

## 5 CONCLUSIONS

We have shown that simple data augmentation operations can boost performance on text classification tasks. Although improvement is at times marginal, EDA substantially boosts performance and reduces overfitting when training on smaller datasets. Continued work on this topic could include exploring the theoretical underpinning of the EDA operations. We hope that EDA's simplicity makes a compelling case for its widespread use in NLP.

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

## 6 APPENDIX

This section contains implementation details, dataset statistics, and detailed results not included in the main text.

### 6.1 IMPLEMENTATION DETAILS

All code for EDA and the experiments in this paper will be made available. The following implementation details were omitted from the main text:

**Synonym thesaurus.** All synonyms for synonym replacements and random insertions were generated using WordNet (Miller, 1995). We suspect that EDA will work with any thesaurus.

**Word embeddings.** We use 300-dimensional Common-Crawl word embeddings trained using GloVe (Pennington et al., 2014). We suspect that EDA will work with any pre-trained word embeddings.

**CNN.** We use the following architecture: input layer, 1-D convolutional layer of 128 filters of size 5, global 1D max pool layer, dense layer of 20 hidden units with ReLU activation function, softmax output layer. We initialize this network with random normal weights and train against the categorical cross-entropy loss function with the adam optimizer. We use early stopping with a patience of 3 epochs.

**RNN.** The architecture used in this paper is as follows: input layer, bi-directional hidden layer with 64 LSTM cells, dropout layer with $p$=0.5, bi-directional layer of 32 LSTM cells, dropout layer with $p$=0.5, dense layer of 20 hidden units with ReLU activation, softmax output layer. We initialize this network with random normal weights and train against the categorical cross-entropy loss function with the adam optimizer. We use early stopping with a patience of 3 epochs.

### 6.2 BENCHMARK DATASETS

Summary statistics for the five datasets used are shown in Table 3.

| **Data** | $c$ | $l$ | $N_{train}$ | $N_{test}$ | $|V|$ |
|----------|-----|-----|-------------|------------|-------|
| SST-2 | 2 | 17 | 7,447 | 1,752 | 15,708 |
| CR | 2 | 18 | 4,082 | 452 | 6,386 |
| SUBJ | 2 | 21 | 9,000 | 1,000 | 22,329 |
| TREC | 6 | 9 | 5,452 | 500 | 8,263 |
| PC | 2 | 7 | 39,418 | 4,508 | 11,518 |

Table 3: Summary statistics for five text classification datasets. $c$: number of classes. $l$: average sentence length (number of words). $N_{train}$: number of training samples. $N_{test}$: number of testing samples. $|V|$: size of vocabulary.

### 6.3 TRAINING SET SIZING

In Figure 4, we show performance on individual text classification tasks for both normal training and training with EDA, with respect to percent of dataset used for training.

### 6.4 EASE OF USE

In Figure 4, we compare the EDA's ease of use to that of related work.

---

[1]Hu et al. (2017) for text classification
[2]Kobayashi (2018) on text classification

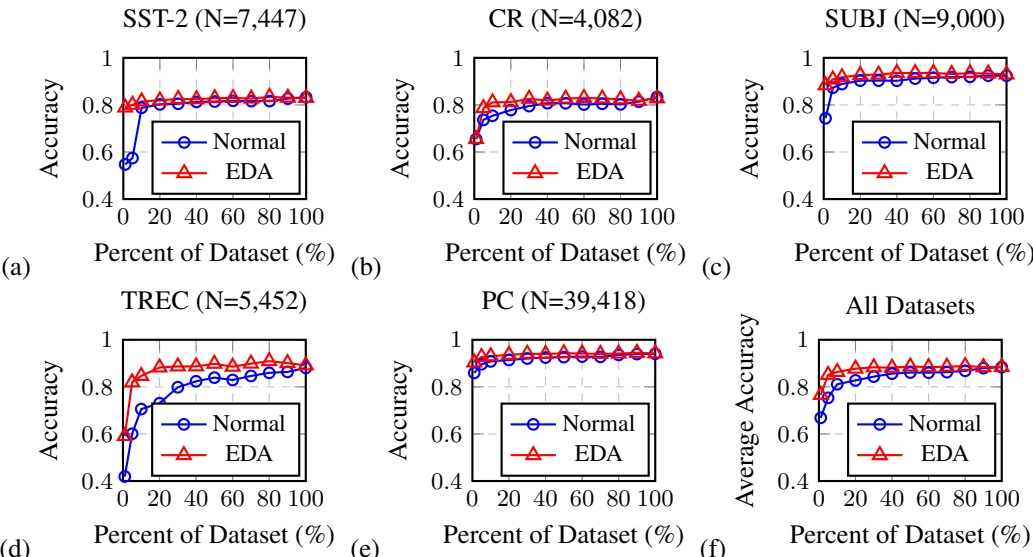

Figure 4: Performance on text classification tasks with respect to percent of dataset used for training.

| Technique (#datasets) | Gain | LM | Ex Dat |
|---|---|---|---|
| VAE+discrim.[1] (2) | ~3% | yes | yes |
| Contextual aug.[2] (5) | 0.5% | yes | no |
| **EDA (ours) (5)** | **0.8%** | **no** | **no** |

Table 4: Related work in data augmentation. #datasets: number of datasets used for evaluation. Gain: reported performance gain on all evaluation datasets. LM: requires training a language model or deep learning. Ex Dat: requires an external dataset.

# 7 FREQUENTLY ASKED QUESTIONS

***How* does using EDA improve text classification performance?** While it is hard to identify exactly how EDA improves the performance of classifiers, we believe there are two main reasons. The first is that generating augmented data similar to original data introduces some degree of noise that helps prevent overfitting. The second is that using EDA can introduce new vocabulary through the synonym replacement and random insertion operations, allowing models to generalize to words in the test set that were not in the training set. Both these effects are more pronounced for smaller datasets.

**Why should I use EDA instead of other techniques such as contextual augmentation, noising, GAN, or back-translation?** All of the above are valid techniques for data augmentation, and we encourage you to try them, as they may actually work better than EDA, depending on the dataset. But because these techniques require the use of a deep learning model in itself to generate augmented sentences, there is often a high cost of implementing these techniques relative to the expected performance gain. With EDA, we aim to provide a set of simple techniques that are generalizable to a range of NLP tasks.

**Is there a chance that using EDA will actually hurt my performance?** Considering our results across five classification tasks, it's unlikely but there's always a chance. It's possible that one of the EDA operations can change the class of some augmented sentences and create mislabeled data. But even so, "deep learning is robust to massive label noise" (Rolnick et al., 2017).

**For random insertions, why do you only insert words that are synonyms, as opposed to inserting any random words?** Data augmentation operations should not change the true label of a sentence, as that would introduce unnecessary noise into the data. Inserting a synonym of a word

in a sentence, opposed to a random word, is more likely to be relevant to the context and retain the original label of the sentence.

