# OpenReview forum: "EDA: Easy Data Augmentation Techniques for Boosting Performance on Text Classification Tasks"
_ICLR.cc/2019/Workshop/LLD — LLD 2019_

### Official Review · AnonReviewer2 · 2019-03-30
**Simple data augmentation techniques for text that work well especially on small datasets**

**Rating:** 4
**Confidence:** 2

**Review:**

Summary: The paper proposes 4 data augmentation techniques for text: synonym replacement, random insertion, random swap, and random deletion. These techniques are validated on 5 text classification datasets, showing increased performance of up to 3% when the dataset size is small. Ablation studies show that all four techniques can be helpful, and suggest sensible values of hyperparameters to set.

Strengths:
1. The proposed techniques are simple and easily implemented. There's no need to train complicated language models.
2. The careful ablation studies (Section 3.3 and 3.4) reveals insights about how much these techniques help and in which context.
3. Situating the proposed scheme in the context of data augmentation for text (Section 4 and 7) is very helpful.

Minor point: the name EDA (easy data augmentation) might be too generic.

A reference for the authors: A related technique for data augmentation for text is called data recombination:
Robin Jia and Percy Liang. Data Recombination for Neural Semantic Parsing. ACL 2016.

---

### Official Review · AnonReviewer1 · 2019-04-06
**A nice clear paper for text-related fast data augmentation techniques**

**Rating:** 3
**Confidence:** 3

**Review:**

The authors present a framework of fast and easy methods for boosting text classification. The methods include synonym replacement, random insertion of a word, random swap of two words and last random deletion. They empirically prove that their approach can increase accuracy, especially in small training set sizes.

The writing of the paper was very clear and easy to understand. The authors had a very extensive section with experiments, analysis, ablation studies as well as comparison to related previous work. The actual contributions of the paper are random insertions, swaps, and deletions as synonym replacement was previously

Pros:
- fast and easy methods
- compared with training techniques

Cons:
- no learning or training
- theoretical explanation

I really liked the frequently asked questions section in the appendix, where the authors respond to questions that easily arise.

Because the gains are indeed marginal, statistical significance is something that should be added. As the authors state in the conclusion, a small theoretical explanation of the EDA operations could be done here, as the methods themselves did not include something extremely complex. The methods should be first compared with a non-training or learning approach, for example adding knn words as augmentation.

Some questions that could be answered as well are: did you explore which replacements affected the model more? for example verbs, nouns or what was their POS-tag? Did you observe any pattern concerning that? As the random deletion seems to be the most effective for small training set sizes, what were the words that were deleted and what can you comment about this phenomenon?

---

### Decision · Program_Chairs · 2019-04-08
**Acceptance Decision**

Accept